# A Study on Internet News for Patient Safety Campaigns: Focusing on Text Network Analysis and Topic Modeling

**DOI:** 10.3390/healthcare12191914

**Published:** 2024-09-24

**Authors:** Sun-Hwa Shin, On-Jeon Baek

**Affiliations:** Nursing Department, College of Nursing, Sahmyook University, Seoul 01795, Republic of Korea; shinsh@syu.ac.kr

**Keywords:** patient safety, hospitals, safety management, hand hygiene, patient-centered care

## Abstract

*Background/Objectives:* This study aimed to identify the main issues related to public patient safety campaigns reflected in Korean online news. This study utilized a text-mining method to identify keywords and topics related to patient safety campaigns. *Methods:* The data collection period was from 1 January 2022 to 31 December 2023, and 4110 news articles were extracted. Through data preprocessing, 2661 duplicated news and 1213 unrelated news were removed, and 236 news were selected. Using the NetMiner program, keyword co-occurrence frequency calculation, keyword centrality analysis, and topic modeling analysis were performed. *Results:* The results showed that the most frequently mentioned keywords with high degree centrality, betweenness centrality, and closeness centrality in online news were “hospital”, “medical”, “medicine”, “project”, and “treatment”. The topics of online news related to the patient safety campaign were “patient-centered care for medical safety”, “health promotion projects at a regional institution”, “hand hygiene education to prevent infection”, “healthcare quality improvement through the Mint Festival”, and “safe use of medicines”. *Conclusions:* This study analyzed patient safety campaign news topics using text network analysis and topic modeling. It was confirmed that patient safety campaigns are essential for fostering a patient safety culture, improving medical quality, and encouraging patient participation in hospitals. Therefore, to build a safe medical environment, it is necessary to establish an effective patient safety campaign for not only medical staff providing medical care, but also patients and their caregivers, and for this, cooperation and participation from various professional occupations are necessary.

## 1. Introduction

The process of receiving a cure in a medical institution involves various factors that can affect a patient’s health and recovery. Particularly, policies and activities for patient safety conducted by medical institutions are crucial in determining the quality of healthcare [1]. Furthermore, improving patient safety in medical services is becoming an important public health issue [2]. The World Health Organization (WHO) defines ‘patient safety’ as reducing all adverse events that occur due to the failure to provide appropriate medical services at an appropriate time [3]. Although the importance of activities for patient safety is increasingly emphasized, many patient safety incidents still occur in medical institutions [4]. Unsafe medical services hinder patient recovery and cause financial losses owing to increased medical costs [1,3]. Therefore, the topic of patient safety is a critical concern in the medical field.

Although there are differences in several indicators when comparing Korea’s patient safety problems with other countries, Korea ranked relatively high among countries of the Organization for Economic Co-operation and Development (OECD) in terms of patient safety [5]. On the other hand, developed countries such as the United States and the United Kingdom ranked somewhat lower with respect to patient safety, which was analyzed to be related to high costs and low prevention rates associated with medical accidents [5]. In Korea, the report rate of voluntary patient safety incidents has been increasing since 2016, and according to the patient safety reporting learning system report, falls and medication incidents accounted for a large proportion [6,7]. In this way, Korea is continuously working to strengthen the data collection and reporting system related to patient safety incidents.

Many studies on patient safety have focused on healthcare providers, but recently, various studies have been conducted on the topic of patient safety, including patients who are the main parties concerned [8,9]. From the patient’s perspective, patient safety means a state in which one’s safety is not threatened [10]. Therefore, patients’ active participation in patient safety activities is the best way to prevent safety incidents [11]. Accordingly, support is being provided to encourage patients’ active participation in patient safety [3,12]. As a strategy to improve patient safety participation, collaboration between medical staff and patients is essential [9], and it is necessary to devise measures to increase positive patient experience and patient safety awareness [12].

The World Health Organization has emphasized the need for international efforts to improve patient safety in developing countries since 2002. It has been conducting extensive campaigns ranging from hand hygiene practices, fall prevention management, surgical safety management, and medication management [3]. In this way, promotions and campaigns are underway to encourage active participation among patients, including medical personnel, to improve patient safety [13,14]. The “Good Catch Campaign” to improve patient safety before and after surgery effectively raised awareness of patient safety by improving scores such as openness of communication between medical staff, feedback on errors, and frequency of incident reporting [14].

In Korea, following the recommendation of the Organization for Economic Cooperation and Development [15] that the problem of the healthcare system is the lack of a clear mechanism to ensure patient safety, various attempts are being made to provide quality assessment and value-based approaches and patient-centered healthcare [16]. In line with these changes, South Korea is organizing the “Mint Festival” to raise awareness of patient safety among patients and caregivers, which is jointly organized by the Korean Hospital Association and the Korean Society for the Improvement of Medical Quality, and in 2023, 110 hospitals nationwide participated [17,18]. “Mint” symbolizes safety [18] as patient safety incidents are associated with red (blood warning) and the mint color is complementary to red. To promote safety activities and effective publicity for patients, medical institutions and the government are holding various patient safety campaign activities, but there has been little research on the core topics and characteristics of patient safety campaigns targeting the general public. To develop effective strategies for patient safety campaigns, it is thought necessary to check how the general public accesses patient safety campaigns and what activities they actively participate in.

As social media and web use increases, the amount of unstructured data generated online exponentially increases. As the Internet and smartphones occupy an increasing proportion of daily life, most people access news through the Internet [19]. In particular, Korea, which has the highest Internet penetration rate in the world, has reported that more than 50% of news users use online news [20]. News can be an appropriate source of information for understanding the main points of issues by focusing on various issues in society [21], and it is necessary to examine what topics are included in patient safety campaigns and how they are delivered. Traditional surveys use structured questionnaires, which have limitations in reflecting the respondents’ various opinions, and to overcome these limitations, studies using big data analysis are increasing [22]. Some studies have reported that analyzing digital news or posts posted on the Internet can represent the actual opinions of the public [23,24].

In the field of nursing, research using online big data has been actively conducted on key issues related to nursing and health [19,23,24,25]. Valuable knowledge can be extracted based on the vast number of data generated online. Previous studies have analyzed social issues related to patient safety searched on the Naver portal site and derived eight themes, such as medical practice, medical personnel, infection and facility, integrated nursing care services, and pharmaceuticals [19]. To expand the understanding of patient safety, which is changing after the enactment of the Patient Safety Act, it is necessary to check what contents are covered in the patient safety campaigns promoted to the general public. Research utilizing online big data can contribute to filling the gap in existing research that focuses on interventions by healthcare professionals by examining key concepts among the general public. Furthermore, the value of this study is to enrich our understanding of patient safety communication from a Korean perspective.

The purpose of this study is to analyze the public’s perception of patient safety campaigns as reflected in Korean online news. To this end, text mining methods were utilized to identify keywords and topics related to patient safety campaigns. The specific objectives are as follows: (1) to identify the frequency and weight of important keywords, (2) to identify the structure and characteristics of the generated network, (3) to identify the network and social connections, and (4) to identify similar clusters of keywords (topics) by constructing semantic connections.

## 2. Methods

### 2.1. Research Design

This was a quantitative content study that utilized online news to explore keywords and research topics by building a network based on the co-occurrence of keywords in articles about patient safety campaigns.

### 2.2. Research Subjects

The subjects of this study were articles on patient safety campaigns posted on Korean online news from 1 January 2022 to 31 December 2023. The selection criteria for online news were limited to news articles with text posted on Naver and Daum, which are representative portal sites with many users in Korea. The exclusion criteria were photo and video news without text, examples of patient safety campaigns in foreign countries, and examples of health-related campaigns that are not related to patient safety (e.g., blood donation campaigns, pharmaceutical company promotional campaigns, police safety campaigns, mountain safety campaigns, fire prevention campaigns, dementia prevention campaigns, food poisoning prevention campaigns, etc.).

### 2.3. Data-Collection Procedure

This study was approved for exemption from review by the S University Institutional Review Board (IRB No. SYU 2023-12-005). It was conducted by searching for the keyword “patient safety campaign” in news on Internet channels (Naver, Daum) and collecting, preprocessing, and analyzing data based on the contents of related articles. Unstructured data collection was performed using Textom 5.0 (The IMC Inc., Daegu, Republic of Korea), and data collection was performed to form the dataset. The data-collection period was set from 1 January 2022 to 31 December 2023. Since words with the same meaning are recognized as different words if they are spaced differently [26], we collected data by entering the following search terms when combining words about patient safety campaigns. The search terms were “patient safety” + “campaign”, “patient-safety” + “campaign”, “patient safety campaign”, and “patient-safety campaign”. After reviewing the contents of online news, articles posted on broadcasting stations, newspaper companies, hospital websites, and forums were selected, and articles on blogs and cafes were mostly personal opinions, so they were excluded from the channel settings.

After going through the process of excluding duplicate data using the editing function, we organized the extracted data in an Excel(Vesion 16.0; Microsoft office, Washington, DC, USA) spreadsheet and confirmed 4110 news items. Data preprocessing involved one researcher reviewing the data organized in an Excel spreadsheet and first removing 2661 duplicate articles by reviewing news titles and link addresses. Afterward, two researchers cross-reviewed the original texts of the articles and excluded 1213 that were not related to this study. The researchers discussed and agreed upon news that was unclear about whether they met the exclusion criteria, ultimately selecting 236 news articles (Figure 1).

### 2.4. Data Analysis

After completing data collection and preprocessing, we used NetMiner (Version 4.5.1.c; Cyram Inc., Gyeonggi-do, Republic of Korea) software for analysis. After extracting morphemes, word refinement was performed using the extraction results. We calculated the frequency of co-occurrence between keywords and created a matrix between words. We analyzed the centrality of keywords in the network and selected core keywords; furthermore, we explored the changing trend of individual core keywords constructed through topic modeling analysis to name topics that reflect the subject. We visualized the constructed matrix, as well as the upper words to appear on the topic-keyword map.

#### 2.4.1. Data Preprocessing

Data preprocessing is the process of extracting morphemes from unstructured text. In this study, we only extracted and analyzed “nouns” among parts of speech, so that the main concepts included in online news could be well understood. First, to analyze the smallest unit of morpheme that has meaning, we repeatedly performed the morpheme separation task and created a dictionary of thesaurus, exception, and defined words [27,28]. Thesaurus contain words that have the same or similar meaning but are written differently depending on the article and are designated as a single representative word. Words such as “hand hygiene”, “hand washing”, and “handwashing” have different spacing but the same meaning; thus, “hand hygiene” was designated as a representative word. In this study, 31 words were registered in a thesaurus. Exception words are words that have no meaning and are irrelevant to the article and should be excluded from the analysis, and we registered 39 of them in the dictionary. Since this study was an article related to a patient safety campaign, we excluded “patient safety” and “campaign” from the analysis. Defined words are determined when two or more words should be read as one word and 81 words were registered as such. We extracted “near miss” as a single word without separating “near” and “miss”. All processes were carried out by one research team member and one nursing professor, and the final decision was made after confirmation and discussion by the entire research team.

#### 2.4.2. Network Analysis and Visualization

For network generation targeting the extracted words, a network was created to identify the relationship between words that appeared closely using the distance information between words. We selected sentences as the unit of co-occurrence in online news and included cases where two words were written consecutively, or where other words were connected between two words. Regardless of the order in which words appeared, extracted all cases where words appeared simultaneously more than twice. Additionally, as it is difficult to understand the phenomenon when the number of co-occurring sentences (links) is too large, this study visualized the top 30 networks with the highest co-occurrence frequency, considering the interpretation and visualization of the research results. Afterward, the two-mode network of word-sentence was analyzed and converted into the one-mode network of word–word.

#### 2.4.3. Centrality Analysis of Keywords

This study performed a centrality analysis on the generated network to identify influential words in the network. This study utilized degree centrality, betweenness centrality, and closeness centrality, and based on previous research, the centrality value was expressed as a value between 0 and 1 to enable comparison between words [28]. The value increases as a word is connected to many other words and is used in various ways in the network, and it is recognized as a core word based on this value. Betweenness centrality is a concept that measures whether a word is located between multiple word groups and plays a bridging role by connecting different topics. Words with high values for degree centrality, betweenness centrality, and closeness centrality are considered core words in text network analysis. In this study, we extracted and analyzed the top 30 words in degree centrality, betweenness centrality, and closeness centrality.

#### 2.4.4. Topic Modeling Analysis

Topic modeling is a statistical method that can find specific topics included in sentences of online news and be analyzed using the Latent Dirichlet Allocation (LDA) algorithm. The LAD algorithm calculates the probability distribution of specific words appearing in a specific sentence to identify a topic’s potential tendency, and the closer the silhouette coefficient value is to “1”, the better the clustering [19]. In this study, we used k-means clustering to calculate the silhouette coefficient for selecting the number of topics. At this time, we used the Monte Carlo Markov Chain (MCMC) as the input option of LDA. To extract topics, we repeatedly performed the process of changing the range of α, β, and the number of topics. Afterward, we checked the top 20 topic models by silhouette coefficient and selected the topic modeling in which the topic words that appeared in the topic model were exclusively classified while being included in the top 10 by silhouette coefficient. We named the topic of the topic group by referring to the top words for each topic. Moreover, using the two-mode network of topic words, we visualized the upper words appearing in the topic–keyword map.

## 3. Results

### 3.1. Word Extraction and Filtering

In this study, after registering similar, designated, and excluded words, we performed data preprocessing on 236 online news articles, and we extracted 3925 words through morphological analysis. As a result of checking the frequency of word appearance, words with a frequency of one time accounted for 39.3% of all words; words with a frequency of 10 or more times accounted for 16.6%, and words with a frequency of 20 or more times accounted for 8.4% (331 words). In this study, words with a frequency of 20 or more were selected to derive meaningful results. Considering the characteristics of the Korean language, single-letter words that are difficult to understand the meaning of; demonstrative pronouns such as this, that, and it; and negative words were excluded from the selection. Finally, 313 words were extracted, and the frequency of words in sentences was 18,456.

After examining the simple frequency and relevance ranking of words used in online news about patient safety campaigns, the top 30 words with the highest simple frequency of occurrence were presented along with the frequency (Table 1). The top words with high simple occurrence frequency were “hospital”, “medical”, “project”, “medicine”, and “institution”.

### 3.2. Network Analysis and Visualization

In this study, we created a network based on the criteria of co-occurrence frequency of three or more times and a distance between words within two words. As a result of creation, we found 3027 links, a network density of 0.04, an average degree of connection of 18.14, an average connection distance of 2.24, and a maximum direct hit of 5.00. Based on the analysis results, one keyword co-occurred with an average of 18 other keywords and went through an average of 2.2 steps. Additionally, all keywords were connected within four steps. As a result of network analysis, we confirmed that a large group was formed centered on words such as “hospital”, “medical”, and “medicine” (Figure 2).

### 3.3. Analysis of the Centrality of Keywords

After checking the keyword centrality of the 313 extracted words, we checked the degree centrality, betweenness centrality, and closeness centrality. The top 30 words with high centrality and the results were presented (Table 1). Words that were included in the ranking due to their high centrality but low frequency of appearance were “system”, “relevance”, and “evaluation”. Additionally, words excluded from the ranking owing to their high frequency of appearance but low centrality were “caregiver”, “effort”, and “hold”.

### 3.4. Topic Modeling

We calculated the number of topics in online news related to the patient safety campaign through the silhouette coefficient using k-means clustering, and the coefficient was calculated using the default settings of the NetMiner program. To select the optimal number of topics, α was set to 0.1–0.2; β was set to 0.01–0.02; iteration was set to 1000; and 136 combinations were identified. After confirming the distribution of words corresponding to each topic in the top 10 silhouette coefficients among the 136 combinations, we reviewed the validity of the selection of the number of topics. We selected five topic groups as input options for LDA (α = 0.16, β = 0.02, and Iteration = 1000), which were judged to be exclusively divided as the silhouette coefficient calculation values were in the top 5. The frequency of the five topics was checked and presented in Figure 3.

As a result of topic modeling, five topics were extracted and identified as the top five keywords (Table 2). Afterward, we reviewed the main text of online news containing the keywords and named the topics as follows.

Topic 1 accounted for 22.7% of all topics. The core keywords were “treatment”, “hospital”, “medical”, “evaluation”, and “institution”. This topic included safety education for hospitalized patients, patient participation in the treatment process, and hospitals where patients can receive treatment with peace of mind. Topic 1 was named “patient-centered care for medical safety”.

Topic 2 accounted for 20.2% of all topics. The keywords were “medical”, “hospital”, “health”, “institution”, and “region”. The topics included quality improvement for community health promotion, the linking of regional medical institutions to increase access to medical care, and the role of regional hub hospitals. Topic 2 was named “health promotion projects at a regional institution”.

Topic 3 accounted for 18.0% of all topics. The keywords were “management”, “prevention”, “infection”, “education”, and “hand hygiene”. The topic included content on infection prevention activities for patient safety, the importance of infection control, and preventive measures to prevent the spread of infection. Topic 3 was named “hand hygiene education to prevent infection”.

Topic 4 accounted for 23.3% of all topics and was the most relevant topic. The keywords were “project”, “hospital”, “progress”, “improvement”, and “mint”. The topic included content such as making mint trees and reflecting on the meaning of patient safety through Mint Festivals. Topic 4 was named “healthcare quality improvement through the Mint Festival”.

Topic 5 accounted for 15.8% of all topics. The keywords were “medicine”, “institution”, “usage”, “side effect”, and “region”. The topic included contents such as drug safety campaigns, creating a culture of safe drug use through collecting and analyzing drug side effects, and safe use of drugs. Topic 5 was named “safe use of medicines”.

We visualized additional keywords, including five keywords for each of the five topics, as a network map in Figure 4.

## 4. Discussion

This study was able to confirm how patient safety activities are carried out and how patient safety promotion is carried out in Korea by identifying various topics related to patient safety campaigns. Based on this, it provided basic data that can be effectively utilized for research on patient safety for not only patients visiting medical institutions but also the general public. Based on the results of this study, we present the following discussion.

The keywords that appeared most frequently in online news related to patient safety campaigns and had a high degree of connection centrality, betweenness centrality, and closeness centrality between words were “hospital”, “medical”, “medicine”, “treatment”, and “project”. These results showed that articles related to patient safety campaigns were centered on hospitals; patients were the main target, and safety activities performed in the process of delivering medical services were the main content. As a result of analyzing topic modeling, topic 1 was “patient-centered care for medical safety”; topic 2 was “health promotion projects at a regional institution”; topic 4 was “healthcare quality improvement through the Mint Festival”; and the three topics were connected by the keyword “hospital”. These results showed that patient safety campaigns are being conducted mainly in hospitals. Currently, various studies are being conducted on the topic of patient safety, but only a few studies focus on patient safety campaigns in hospitals [14]. Various research approaches are needed to share information on patient safety quality management and campaigns being held in hospitals. Additionally, patient safety campaigns mainly conducted in large hospitals should be expanded to small- and medium-sized hospitals and primary medical institutions to create a safe medical environment.

In this study, topic 1 was “patient-centered care for medical safety”. This included building a partnership between medical staff, patients, and their caregivers, respecting the patient’s wishes, needs, and preferences, and involving patients in curative decision making. Patient-centeredness means that medical staff support patients and their caregivers so that patients are the center of the patient–medical staff relationship and the values that patients consider important can be reflected [29]. Since the 2000s, many countries have considered patient-centeredness as an essential part of quality assessment and performance evaluation of the public health care system [30]. In Korea, “patient experience assessment” has been conducted on patients since 2017 to realize patient-centered care and improve the quality of care [16]. Previous studies have suggested the participation of patients and their caregivers in the cure process to improve patient safety and quality of care [30,31]. Many previous studies have proven that patient participation has a positive effect on preventing safety incidents [8,32]; the U.S. Agency for Healthcare Research and Quality has implemented 20 tips for error prevention and 10 questions for patients and guardians and The Joint Commission has been educating about the risks of safety incidents and emphasizing patient responsibility in all processes of medical services by implementing activities such as the Speak up campaign and Take CHARGE [33,34]. It was suggested that such patient and guardian participation can reduce adverse events and produce positive outcomes, and the need for an intervention program to expand patient and guardian participation was emphasized. [35,36]. To receive safe medical services, patients themselves should play an active role in patient safety activities. Moreover, to implement patient-centered medical services in the medical environment, it is necessary to establish a system to enable smooth information sharing and communication between medical staff and patients.

Topic 2 was “health promotion projects at a regional institution”. This included quality improvement for promoting community health and the linking of regional medical institutions to increase medical accessibility and medical services for community care. Regional public hospitals are promoting various services such as maternal and child health, infectious diseases, and emergency care to provide high-quality medical services to local residents [37]. In particular, hospitals located in areas with low population density play a critical role as healthcare institutions in the region [38]. Nevertheless, regional hospitals located in small- and medium-sized cities and counties in local areas still have difficulty securing medical personnel and cannot faithfully fulfill their role in providing comprehensive health care and disease prevention to local residents [37,39]. Accordingly, the Korean government is trying to establish the role of regional hospitals by expanding the scope of the service to include integrated community care, primary care home visits, and home care pilot projects [39]. In a previous study, Korean news articles were analyzed with integrated community care as a keyword; leading projects and the community were classified into five topics including return-linked cooperation, medical issues, and expert opinions [40]. Specific proposals to resolve regional medical inequalities in Korea, such as the activation of home visits and home medical care, can be expected to play a complementary role in terms of continuity of treatment and cost. Therefore, it is necessary to establish an integrated medical system by linking with the regional institution and not limiting the patient safety improvement campaign to hospitals. In addition, conducting various forms of campaigns at community medical institutions will contribute to raising awareness of patient safety among the general public.

Topic 3 was “hand hygiene education to prevent infection”. It covered proper hand washing, hand hygiene education, and the importance of personal hygiene to prevent infectious diseases. Despite global efforts to manage healthcare-associated infections, in the United States, approximately 4–5% of hospitalized patients develop healthcare-associated infections, and approximately 75,000 of those infected die annually [41]. In particular, hand hygiene is the most basic and important behavior to prevent healthcare-associated infections. Over the past 20 years, various interventions have been implemented for healthcare workers to practice proper hand hygiene to prevent healthcare-associated infections, and effective hand washing using alcohol-based hand sanitizers has been recommended [42]. While research on hand hygiene practices for healthcare workers is actively being conducted, studies on hand hygiene practices for hospitalized patients and the general public are lacking. In a previous study that observed the hand hygiene performance rate of 454 caregivers in a liver transplant ward, the highest compliance rate was observed: “after touching the patient”, while the lowest was observed “before touching the patient” and “before washing/aseptic procedures” [43]. Thus, it is necessary to encourage patients, caregivers, and caregivers to actively participate in hand hygiene promotion activities. In another study, patients were less willing to ask general questions related to cure and health than to confirm or request hand washing [44]. This is thought to be due to the cultural environment that makes it difficult for patients to provide or request feedback on the actions of medical staff. Since the COVID-19 pandemic, the importance of hand hygiene has been emphasized to the general public, and various campaigns have been conducted through the media. Thus, it is necessary to emphasize hand hygiene promotion campaigns to prevent healthcare-related infections and conduct empirical research on the proper practice of hand hygiene targeting patients and caregivers to accumulate evidence.

Topic 4 was “healthcare quality improvement through the Mint Festival”. This included in a national campaign to raise awareness of the quality and safety of healthcare among patients and their caregivers, including medical staff. In Korea, the Mint Festival was held in August 2023 to commemorate Patient Safety Day (September 17) designated by the WHO [17,18]. The Mint Festival refers to a campaign to raise national awareness of the quality and safety of healthcare [17,18]. When thinking of patient safety incidents, the color red (blood, warning) comes to mind; thus, the complementary color of red, mint, was used with the meaning of safety, and the Korean word for mint, pakha, was used [18]. The content of the peppermint campaign shows that medical staff and patients/caregivers are participating in various ways, such as by decorating peppermint trees to pledge and confirm patient safety, displaying patient safety activity posters, and displaying examples of patient safety improvement activities. Regarding patient safety, the “Speak-Up” campaign has been conducted for many years, focusing on medical staff, to identify concerns that may affect patient safety [45,46]. Additionally, the “Good Catch Campaign” was effective in improving patient safety through smooth communication between medical staff before surgery and feedback on errors [14]. In Australia, the “Measure-up” campaign was conducted to improve awareness of obesity prevention, and the group that recognized the campaign message well had stronger behavioral changes and intentions regarding obesity than the group that did not [47]. Currently, in hospitals with quality improvement departments, patient safety specialists play a central role in monitoring the level of patient safety activities and making multifaceted efforts to prevent patient safety incidents and improve the quality of medical care. The theme of World Safety Day in 2023 was “Patients’ Active Participation for Patient Safety”, and in line with this, the Mint Festival was held to provide an opportunity to promote “patient safety” not only to medical staff but also to patients and their caregivers. In this way, patient safety should not be considered a task that only medical staff should be responsible for but a campaign that promotes it so that patients also take an interest as stakeholders is required. It is necessary to conduct a large-scale patient safety campaign for patients, their caregivers, and the general public to contribute to raising awareness of patient safety.

Topic 5 was “safe use of medicines”. It covered the use of drugs administered directly to patients and the reporting of side effects. Drugs are essential resources for treating patients, but they can act as a factor threatening patient safety [48]. As a result, errors related to drug administration are the most frequent adverse events occurring in hospitals [49]. Medication errors are a major cause of patient morbidity and mortality and cause serious problems such as enormous medical costs [4,50]. Therefore, when medication errors occur, they should be reported immediately, and appropriate measures should be taken to identify the cause through the report’s content [4,51]. In a previous study conducted on nurses, we reported work intensity, fatigue, and stress levels as causes of medication errors [52,53]. As such, medication errors occur due to incorrect dose, incorrect time, incorrect frequency, incorrect patient identification, omission of medication, and incorrect medication [4,54]. However, medication errors can be prevented through repeated and effective education; thus, it is necessary to conduct periodic medication education for medical professionals in charge of drug administration and establish a system that analyzes various factors that may affect medication errors [55]. Recently, hospitals have been utilizing programs such as mobile medication systems to prevent medication errors. Mobile medication systems display medication error warnings on the point-of-care device screen when there is a discrepancy between the doctor’s prescription and the patient’s information, limiting additional work and storing error logs [56]. It is necessary to promote these mobile medication systems to patients and improve them so that patients can participate in the medication process. Additionally, it is necessary to encourage patient participation by repeatedly providing education related to medication compliance to patients who are the subjects of medication, so that they can confirm and take the correct dosage and administration.

This study was significant because it analyzed a large dataset of online news articles through topic modeling and network analysis to identify complex themes and trends in patient safety campaigns. In addition, by examining public campaigns, this study contributed to filling a gap in existing patient safety research by identifying the direction of media attention to patient safety for the general public rather than focusing on interventions for healthcare professionals.

Nevertheless, it has several limitations. First, the data covered in this study were limited to online news published domestically and thus have limitations. In the future, we suggest big-data analysis research that expands the scope of online news to include articles published overseas and cover major issues in patient safety. Second, the data extracted in this study consisted of online news published by hospitals and medical institutions, and the content was mainly written from the perspective of hospitals hosting the campaign. In the future, we propose a comparative analysis study by adding media sources other than hospitals and extracting data from social media sources written by patients, guardians, and members of the public who have experienced patient safety campaigns. Third, this study had limitations in identifying temporal trends by collecting online news published in 2022 and 2023. In addition, there was a possibility of potential bias in the sampling and data processing processes. Therefore, future studies should extend the data collection period to explore trends in patient safety campaigns over time, use sampling methods such as stratified sampling, and verify inter-researcher agreement to minimize potential bias. Fourth, patient safety campaigns are currently being conducted in various forms, including the “Mint Festival”, targeting patients and caregivers. Therefore, we suggest empirical research that investigates patient safety promotion and campaigns conducted mainly by hospitals and verifies their effectiveness.

## 5. Study Implications

Patient safety campaigns are important for short-term activities such as smoking cessation and vaccination, but it is essential to encourage patient participation in continuous and long-term patient safety strategies, implement patient-centered care in hospitals, and provide continuous care for patients returning to the local community after discharge. Therefore, based on these results, hospitals need to establish systematic programs that can expand the patient safety culture and develop more efficient and effective patient safety education for patients to prevent patient safety incidents that can harm patients. In addition, although patient safety campaigns are currently being carried out mainly in hospitals, it is necessary to expand the target and scope and conduct promotional activities that consider the target so that the general public can take an interest in patient safety, and a comprehensive approach is needed so that the target can directly participate in patient safety activities when they visit the hospital.

## 6. Conclusions

This study examined how patient safety campaign-related content was reported and delivered to the public through online news. The research results showed that the major news topics were “patient-centered care for medical safety”, “health promotion projects at a regional institution”, “hand hygiene education to prevent infection”, “healthcare quality improvement through the Mint Festival”, “practicing hand hygiene to prevent infection,” and “safe use of medicines”. By examining various topics related to patient safety campaigns, we determined how activities and promotion of patient safety were conducted. Through this, we found that patient safety campaigns are an essential element for creating a patient safety culture, improving the quality of healthcare, and encouraging patient participation. We propose to conduct continuous research on future patient safety campaigns to develop and evaluate effective patient safety campaigns targeting not only medical staff but also patients and their caregivers. Above all, we should take the lead in creating a healthy patient safety culture by systematizing financial support strategies along with improving the legal system for patient safety at the national level and expanding it to not only large hospitals but also small- and medium-sized hospitals and primary medical institutions.

## Figures and Tables

**Figure 1 healthcare-12-01914-f001:**
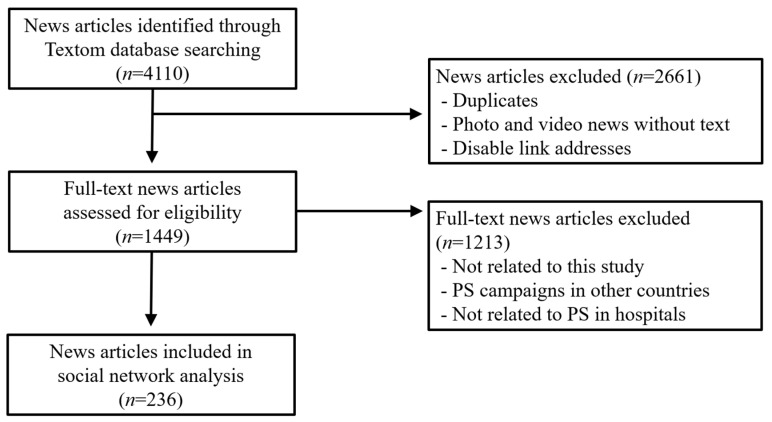
Preferred reporting items for systematic reviews and meta-analyses (PRISMA) flow diagram of the online news selection.

**Figure 2 healthcare-12-01914-f002:**
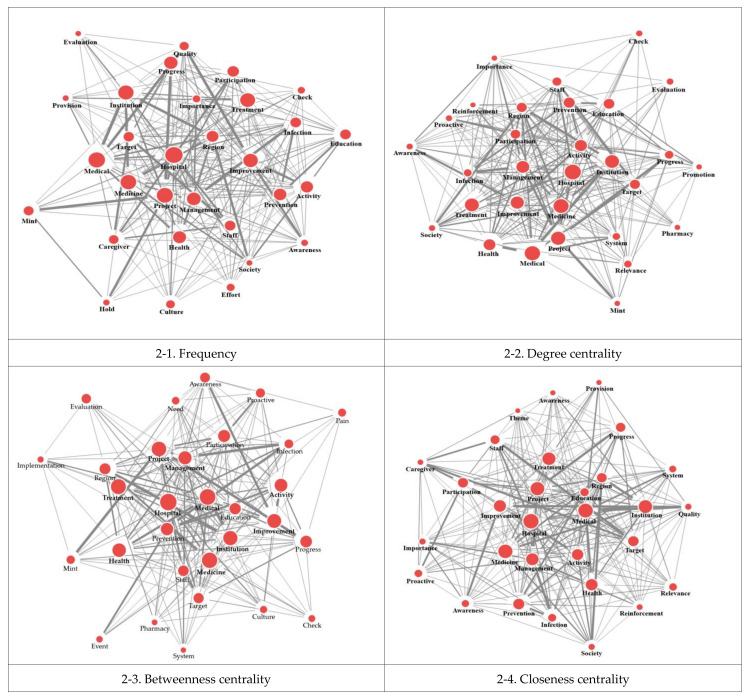
Spring network map of centrality.

**Figure 3 healthcare-12-01914-f003:**
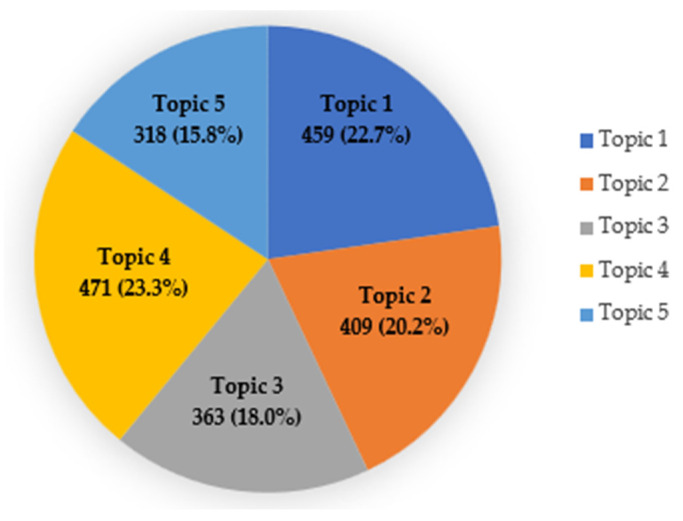
Distribution by topic.

**Figure 4 healthcare-12-01914-f004:**
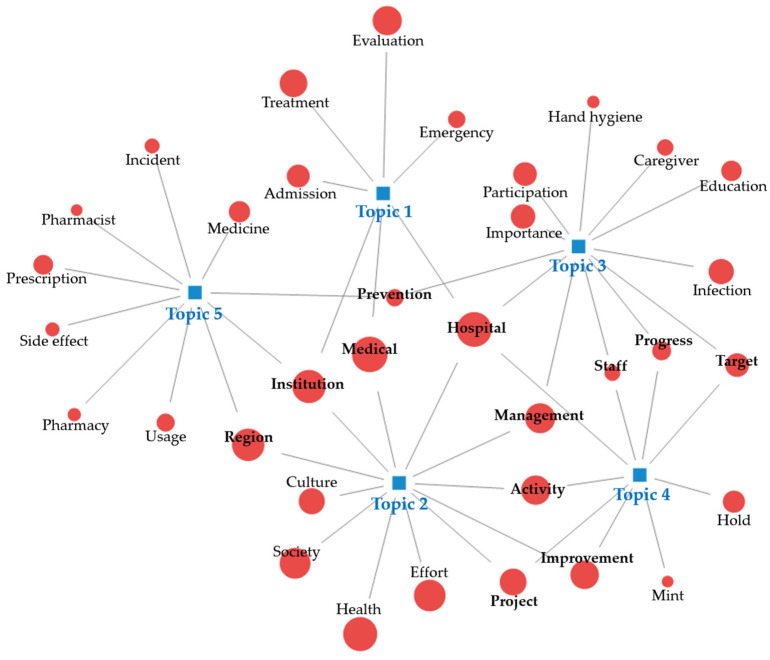
Topic group network by topic modeling.

**Table 1 healthcare-12-01914-t001:** Top 30 keywords by frequency and centrality analysis.

Rank	Keyword							
	Frequency		Degree Centrality		Betweenness Centrality		Closeness Centrality	
1	Hospital	809	Hospital	0.440	Hospital	0.212	Hospital	0.634
2	Medical	490	Medical	0.309	Medical	0.091	Medical	0.586
3	Project	466	Medicine	0.270	Medicine	0.080	Medicine	0.565
4	Medicine	451	Project	0.263	Treatment	0.072	Project	0.564
5	Institution	384	Treatment	0.251	Project	0.066	Institution	0.561
6	Treatment	337	Institution	0.251	Institution	0.061	Treatment	0.558
7	Improvement	313	Improvement	0.238	Improvement	0.048	Management	0.555
8	Management	311	Management	0.226	Health	0.045	Improvement	0.554
9	Progress	291	Activity	0.202	Management	0.040	Health	0.543
10	Health	271	Health	0.200	Activity	0.034	Activity	0.539
11	Prevention	238	Prevention	0.170	Prevention	0.025	Prevention	0.532
12	Activity	232	Education	0.168	Participation	0.024	Target	0.529
13	Region	197	Region	0.158	Progress	0.024	Participation	0.526
14	Participation	194	Target	0.158	Education	0.024	Region	0.524
15	Staff	192	Participation	0.156	Region	0.023	Progress	0.519
16	Education	192	Staff	0.146	Staff	0.019	Staff	0.516
17	Infection	192	Progress	0.144	Target	0.016	Education	0.512
18	Target	189	Infection	0.124	Awareness	0.012	Infection	0.507
19	Mint	161	System	0.100	Evaluation	0.012	Proactive	0.501
20	Caregiver	148	Relevance	0.100	Proactive	0.011	Relevance	0.501
21	Quality	137	Evaluation	0.097	Infection	0.011	Awareness	0.498
22	Culture	130	Check	0.092	Mint	0.010	System	0.496
23	Effort	126	Proactive	0.092	Pain	0.010	Quality	0.495
24	Importance	125	Awareness	0.092	Needs	0.010	Importance	0.495
25	Check	117	Society	0.092	Culture	0.008	Society	0.495
26	Awareness	116	Promotion	0.088	Check	0.008	Reinforcement	0.493
27	Hold	114	Importance	0.088	Incident	0.008	Caregiver	0.492
28	Provision	113	Pharmacy	0.088	Implementation	0.008	Provision	0.492
29	Society	113	Mint	0.088	Pharmacy	0.008	Operation	0.492
30	Evaluation	107	Theme	0.085	System	0.008	Theme	0.489

**Table 2 healthcare-12-01914-t002:** Topic groups based on topic modeling.

Category	1st Keyword	2nd Keyword	3rd Keyword	4th Keyword	5th Keyword	Topic Group	No of Articles (%)
Topic 1	Treatment	Hospital	Medical	Evaluation	Institution	Patient-centered care for medical safety	459 (22.7)
Topic 2	Medical	Hospital	Health	Institution	Region	Health promotion projects at a regional institution	409 (20.2)
Topic 3	Management	Prevention	Infection	Education	Hand hygiene	Hand hygiene education to prevent infection	363 (18.0)
Topic 4	Project	Hospital	Progression	Improvement	Mint	Healthcare quality improvement through the Mint Festival	471 (23.3)
Topic 5	Medicine	Institution	Usage	Side effect	Region	Safe use of medicines	318 (15.8)
Total							2020 (100.0)

## Data Availability

The datasets used and/or analyzed during the current study are available from the corresponding author upon reasonable request.

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
