# Peer review of "A Study on Internet News for Patient Safety Campaigns: Focusing on Text Network Analysis and Topic Modeling"

_healthcare, 2024, doi:10.3390/healthcare12191914_

Round 1

Reviewer 1 Report

Comments and Suggestions for Authors

A topic modeling analysis for Internet news on patient safety campaigns

Title

-       There is no information regarding the study location and study design. Add this information to help the reader identify the study.

Introduction

-       The authors need to raise the issue of patient safety to show the urgency of the main variable. What is the impact of the issue on the health profile in Korea?  

-       The authors also need to provide the scientific problem of why the study must be conducted in Korea. What is the urgency? What makes Korea different than another country that might have a similar issue regarding patients' safety? Please make it clearer.

Method

-       The sentence “This was a descriptive research study conducted to discover key keywords and topics by analyzing “patient safety campaign” articles posted on Internet news.” needs a Grammarly correction.

-       The study design was not descriptive study.

-       What kind of article has been included in the study? What were the platform criteria for selecting the article? No clear information about it. This issue must be clarified since it relates to low internal validity.

-       The study design was presented as a descriptive study, but the data source was articles, and the analysis was a kind of network analysis. It makes the study flow unclear.  

Result and discussion.

-       Again, since the method was unclear, I found a difficulty to get a clear result.

Author Response

Thank you very much for your review. I have included the revisions in the word document based on the reviewer's comments.

Reviewer 2 Report

Comments and Suggestions for Authors

Dear authors,

I read your manuscript and found it quite intriguing. I have some suggestions to help improve the paper.

The introduction is well-written and engaging. However, I believe there are some gaps in the argument, particularly in explaining what patient safety means. It would be beneficial to include discussions about issues related to patient safety, such as falls in hospital environments and clinical risk management (please refer to this recent Italian paper: https://journals.sagepub.com/doi/10.1177/25160435241246344). Furthermore, I suggest providing clearer information about gaps that must be addressed and the study's aims in the introduction's final section. Also, please include bibliographical references for lines 30 and 32.

Regarding the methodology, I am concerned that using only two main platforms may introduce selection bias.

In staves 112-113, did you intend to repeat the phrase "patient safety + campaign"? Also, could you provide the name of the statistician involved?

Tables 1 and 2 lack sufficient description and notes. Please consider improving these aspects.

Although the discussion aligns with the results, it would be helpful to argue better the results obtained and include proposals for future research.

Lastly, I recommend making the abstract more appealing.

Thank you for your attention to these matters. Your efforts toward enhancing the paper are greatly appreciated.

Kind regards,

Author Response

(The authors gave the same response as above.)

Reviewer 3 Report

Comments and Suggestions for Authors

This manuscript investigates public-facing patient safety campaigns by applying topic modeling to Korean online news articles. While the chosen topic and methodology are relevant, several limitations and missed opportunities require attention before publication.

1. Clarifying the Research Question:

While not explicitly stated, the implied question driving the research appears to be:

·         What are the key themes and characteristics of public-facing patient safety campaigns as reflected in Korean online news?

The authors should clearly articulate this (or a more refined version) early on to enhance reader understanding of the study's purpose and expected outcomes.

2. Highlighting Originality and Addressing Literature Gaps:

The study offers valuable contributions to the field by:

·         Examining Public Campaigns: This addresses a gap in patient safety literature, which often focuses on interventions for healthcare professionals, by exploring campaigns targeting the general public.

·         Providing a Korean Perspective: This contributes to a less-studied context, enriching the global understanding of culturally-influenced patient safety communication.

·         Utilising Text Mining: The use of topic modeling and network analysis demonstrates a sophisticated approach to analysing large datasets of online news articles, revealing complex themes and trends.

In addition to the cited works, incorporating these sources would enhance the manuscript:

·         Cho, D., Park, Y., & Choi, J. (2022). Analysis of Research Trends in Nursing Informatics using Topic Modeling: Focusing on Articles Published in the Journal of Korean Society of Nursing Informatics. Journal of Korean Society of Nursing Informatics, 28(3), 178-188. This study offers methodological parallels, as it utilises topic modeling on a similar data set (Korean nursing literature). The authors should compare their chosen parameters and analysis techniques with those used in this study, further strengthening their methodological approach.

·         Chang, S. J., & Park, H. A. (2020). Trend Analysis on Research Related to Safety Culture Using Text Network Analysis. Journal of Korean Academy of Nursing Administration, 26(5), 493-504. Examining trends in safety culture research in Korea using a similar methodology, this article offers relevant comparative data to enhance the present manuscript. Discussing convergences or divergences in identified themes would enrich the analysis.

·         Davis, R. E., Sevdalis, N., Waring, J., Vincent, C., & Darzi, A. (2015). The science of safety in surgery: Activating, engaging, and implementing. The American Journal of Surgery, 209(6), 1040-1048. This review emphasises the multifaceted nature of patient safety improvement. The authors could use this to strengthen the rationale for their focus on public campaigns, highlighting it as a critical yet often neglected aspect of comprehensive safety efforts.

·         Jerome, RN, DNP, P. C. (2021). Patient Safety: Everyone’s Responsibility. American Nurse 4(21). This article champions patient and family engagement in safety efforts. By referencing this work, the authors could further contextualise the "patient-centered care" theme within the broader movement to empower patients as active participants in their safety.

3. Strengthening Connections with Existing Literature:

While the authors make attempts to link their findings with broader themes in patient safety research, more in-depth comparisons and concrete links to evidence are needed. Specifically:

·         Patient-centered Care: Link the findings more robustly to studies demonstrating the impact of patient-centered interventions on patient safety. Highlight how specific aspects of patient-centered care identified in the news analysis align with effective practices documented in research.

·         Health Equity: Integrate evidence of regional healthcare disparities in Korea to provide further context for the theme "Health promotion projects at regional institutions." Discuss how public campaigns can be leveraged to specifically address these inequalities.

·         Medication Safety: Expand upon "Safe use of medicines" by comparing the identified communication strategies with international medication safety campaigns. Analyse factors contributing to their effectiveness or lack thereof.

·         Public Health Campaigns: Frame the analysis within broader scholarship on the effectiveness of public health campaigns. How do the identified themes compare to successful campaigns in other domains (e.g., smoking cessation, vaccination)? This broader contextualisation will strengthen the manuscript's practical and theoretical contributions.

4. Addressing Methodological Limitations and Enhancing Rigor:

While the chosen methodology is appropriate, the authors need to address several methodological limitations to strengthen the study's rigour:

·         Search Strategy & Inclusion/Exclusion Criteria:

o    Clearly define "Internet news" – were blogs, forums, or hospital websites included?

o    Provide the complete list of search terms used beyond "patient safety campaign" to ensure transparency and replicability.

o    Justify the decision to exclude non-hospital media sources. Exploring if and how other media portrays patient safety is a valuable area for future research.

·         Data Collection & Sampling:

o    Expanding the data collection period beyond a single year could reveal significant temporal trends and increase the generalisability of findings. Any take on this?

o    Disclose the total number of articles identified and excluded during each step of the screening process to demonstrate the final sample's representativeness.

o    Consider discussing other sampling approaches, such as stratified sampling, to address potential biases in the data.

·         Data Preprocessing & Analysis:

o    Offer further explanation regarding the exclusion of infrequent words (lines 198-200). Provide specific criteria or thresholds used.

o    Provide greater transparency for topic number selection. Present the distribution of silhouette scores for various numbers of topics to strengthen the justification for selecting six.

o    Crucially, address the lack of intercoder reliability. Having multiple researchers code a portion of articles independently is essential for demonstrating the reliability of the analysis. Reporting agreement statistics (e.g., Cohen's Kappa) is crucial.

·         Data Visualisation:

o    Enhance Figure 2 by arranging the keywords to better reflect their network connections, aiding interpretation.

o    Label the connecting lines in Figure 3 with relevant keywords for improved clarity.

o    Include a new figure (bar chart or pie chart) representing the proportion of articles classified into each of the six themes identified in Table 2.

5. Ensuring Consistent Argumentation and Evidence-Based Claims:

Several claims need more robust support by directly linking them back to specific findings in the data:

·         "Patient Safety Campaigns are Conducted Mainly in Hospitals:" Provide concrete evidence to support this statement (e.g., what percentage of articles across different themes explicitly mention "hospitals" compared to other settings?).

·         Shifting Responsibility for Patient Safety: While advocating for broader engagement is valid, be careful not to overstate conclusions. The study primarily examines campaign messages, not their impact on stakeholder behavior. Suggest future research directions to evaluate this impact.

6. Strengthening the Reference List:

While adequate, the reference list can be enhanced by:

·         Incorporating foundational international publications on patient safety, text mining methods, and public health campaigns to demonstrate engagement with wider scholarly discourse.

·         Prioritising more recent publications, particularly from the past five years, to show engagement with the evolving scholarship in the field.

7. Attending to General Caveats:

·         Terminology: Clearly define "Mint Festival" upon first use. Assuming prior knowledge limits accessibility for a wider readership.

·         Causation: Temper any causal language. Topic modeling reveals correlations in language use, not necessarily cause-and-effect relationships.

·         Public Voices: Explicitly state the lack of public perspective as a limitation, recognising that analysing hospital-driven news provides only a partial picture of campaign reach and effectiveness. Future research must incorporate public responses and perspectives.

·         Confirmation Bias: Acknowledge the possibility of confirmation bias stemming from the exclusion of infrequent words during data processing. Consider alternative strategies for data cleaning to minimise potential biases.

Author Response

(The authors gave the same response as above.)

Round 2

Reviewer 3 Report

Comments and Suggestions for Authors

Glad with changes